**Perspective**

# Kindness as a public health action
Tri-Long Nguyen [1,2] ✉ & Ji-Young Lee [3]

The current global mental health crisis needs action. Here we show, using empirical evidence of randomised controlled trials combined with Rose's theoretical framework of preventive medicine and epidemiological principles, that the universal practice of kindness is a potentially effective grassroots public health promotion action that could propagate from the individual to the collective. Beyond effectiveness alone, we present medical ethics principles to show that the universal practice of kindness is also relatively costless, socially just, inherently consensual, empowering, and immediately available to, for and within every one of us. We argue that the need for structural changes and future research studies should not justify delaying the individual practice of kindness in our daily lives, since kindness is compatible with research, clinical practice and policy-making.

In 2022, the World Health Organisation (WHO) reported that mental health conditions affected nearly one billion people globally and most alarmingly, one out of seven teenagers worldwide[1]. In parallel, national data from the United States showed a decline in well-being among young adults[2]. Whether this downturn has been triggered by the recent pandemic or other causes, the global mental health crisis affecting youth is undeniable[3]. Advocating for structural changes towards prevention of mental health conditions for those at risk and universal coverage for mental health services[4], the WHO also underscores the importance of autonomy and person-centred approaches to health decision-making[1]. Here, we delineate arguments borrowed from Eastern and Western, modern and ancient, philosophies and sciences, contending that the practice of kindness is a public health action that is effective, costless, socially just, consensual, empowering, and immediately available to, for and within every one of us.

## Borne out of understanding, love, and compassion

Kindness is a volitional endeavour, which does not necessarily require material gift, service or favour; it can be a grand gesture or a tender thought, a greeting or a smile, a "yes" or a "no", to others and to ourselves. We define "Kindness" as loosely designated thought, speech or action intended to benefit others[5], with no expectation of reward or recognition[6]. We relate kindness to the Buddhist notion of *mettā* (Pali term for "benevolence" or "loving kindness"), the practice of which is borne out of cultivating understanding, love, and compassion, first for ourselves and then for others[7]. While one may question how the genuine intention to benefit others can first be practised within ourselves, it seems reasonable to claim that one cannot offer to others what one does not have for oneself. This distinguishes kindness from people-pleasing, which comes from dependency, expects approval from others, and neglects understanding, love and compassion for ourselves[6].

More specifically in the clinical setting, following a recent review by Hake and Post (2023) on the definitions of kindness in healthcare, kindness can be broadly defined as 'an action (mental or physical), that has a *"warm"* component, *"directed toward fostering… well-being or flourishing"*[8]. While Hake and Post suggest defining kindness based on the beneficial result on others[8], we underscore the need to extend the recipients of kindness: kindness towards others (patients and colleagues) must be conditional on kindness towards oneself. We argue that it is fundamental to prevent kindness from being regarded as an "obligation" towards others, as this could result in a source of distress to doctors, thereby being inconsistent with the well-being and flourishing advocated by Hake and Post. Kindness is as much a self-relation as it is an orientation towards others. This argument is akin to the fundamental distinction between empathy and compassion[9,10]. Empathy is a mental construct that allows one to resonate with other's feelings (positive or negative), which can result in a feeling of distress when observing others suffering[9]. It is possible for empathy to manifest as pity or even desensitisation, which entails detachment and passivity[11]. On the other hand, compassion is characterised by a feeling of warmth, concern, and care for the other, as well as a strong motivation to improve the other's wellbeing[9]. Healthcare professionals may be vulnerable to "compassion fatigue", given their proximity to those who are sick, suffering, and so forth[12]. In such cases, exercising self-compassion is a crucial precursor to showing kindness towards others; rather than being overcome with feelings of guilt or self-criticism, self-compassion would involve "acknowledging that suffering, failure, and inadequacies are part of the human condition, and that all people—oneself included—are worthy of compassion[13]." In this sense, we relate kindness to compassion, rather than mere empathy. Thus, we regard kindness as a path towards enhancing resilience of public health rather than merely a social virtue in the moral sense.

[1]Department of Public Health, Section of Epidemiology, University of Copenhagen, Copenhagen, Denmark. [2]Department of Public Health, Copenhagen Health Complexity Center, University of Copenhagen, Copenhagen, Denmark. [3]Department of Public Health, Section for Health Services Research, University of Copenhagen, Copenhagen, Denmark. ✉e-mail: long@sund.ku.dk

## A mass approach to public health

To show how practising kindness can translate into an effective public health action, we use the influential framework of preventive medicine proposed by Rose, who argued that massively applying to the general population an intervention with small individual benefits could be more effective in reducing the number of disease cases than targeting only those at high-risk of disease[14,15]. The rationale behind this "prevention paradox" is simple: although a preventive measure might be more effective in individuals at high risk, the proportion of those high-risk groups is often much lower than the low-risk population, such that targeting only those at high risk would result in preventing a lower number of absolute cases[14,15]. Likewise, we argue that cultivating kindness, understanding, love and compassion, first for oneself, then for others – even if it had only small to moderate individual benefits – is not only for persons at high risk of or affected by mental health conditions, instead, the practice of kindness is for everyone.

Further, assuming the practice of kindness can propagate across interconnected individuals within social networks (e.g., households, circles of friends, communities, etc.), both the exposure to and the effect of kindness could be amplified by a network multiplier, analogous to the reproduction number, $R_0$, in infectious disease epidemiology[16]. In analogy to epidemics, any so-called "individual" behavioural action (e.g., thought, speech, or physical action) could in fact have a collective impact, through the emergence of cascading phenomena[17–20]. The implication is straightforward: whether it is about epidemics or solidarity movements, mass phenomena can emerge from individuals and have an impact on the collective. Examples of kindness and compassion actions emerging from individuals include historical and contemporary mass peace movements around the world in response to violence committed during war.

Taking the critiques raised against Rose's mass approach to public health, we now discuss how the practice of kindness overcomes common clinical, economic, political and ethical dilemmas posed by preventive health strategies imposed on populations.

## Evidence of clinical benefits

### First, do no harm

The principle of non-maleficence is a central tenet of modern medical ethics, providing a standard that public health systems should uphold. Moreover, the principle of beneficence – which requires not only to prevent harm to individuals, but to positively benefit them – further outlines an ethical ideal for the provision of quality healthcare at the individual and population level. Thus, if a public health intervention like practising kindness were harmful in some groups, then a population approach would elevate the risk in those groups and decrease not only the total effectiveness of the strategy[21], but violate the ethical standards of healthcare. Here, existing evidence consistently indicates the benefits and non-maleficence of practising kindness. When kindness takes the form of silent but caring intention, 4 meta-analyses of 22, 21, 19, and 10 randomised controlled trials show that in comparison to no intervention, kindness-based meditation has a moderate effect in reducing depressive symptoms, anxiety and stress, and increases compassion and self-compassion, and positive emotions of kindness-practitioners[22–25]. When kindness translates into action, a meta-analysis of 27 experimental studies shows that practising acts of kindness has a small-to-moderate effect on increasing well-being of the kindness-actors[5]. In other words, empirical evidence shows that – whether it is in the form of caring thoughts or in gestures – practising kindness has at best moderate benefits for those who practise it, and at worst causes no harms, indicating that kindness as an intervention strategy is aligned with the principles of non-maleficence and beneficence. On the other hand, recent data shows that barriers to kindness, such as fear of compassion, are associated with mental health conditions[26]. For instance, during the pandemic, fears of offering compassion to oneself/others and fears of receiving compassion from others were associated with depression, anxiety and stress[27]. In short, empirical evidence about the benefit-to-harm balance consistently indicates a favourable effect of practising kindness; thus, as Asch, Atkins, and Walling (2021) write: "if kindness were a drug, the FDA would approve it"[28].

Given this empirical evidence and following Rose's model, practising kindness – even if the individual benefits were only small and non-communicable – would turn into an effective mass strategy if it were encouraged in all strata of the population. Again, in reality the effect of practising kindness might further be amplified by diffusion across individuals, as kindness might also propagate positive effects on recipients of kind acts[29], which are often underestimated[30], and are perhaps even contagious to spectators of such actions[31].

## A cost-effective approach

A second critical aspect of Rose's model is the constraint posed by intervention costs. In the case where the resources available for a health strategy were limited, applying it massively to the population would raise an economic dilemma against the societal willingness to pay for an intervention that has little individual benefits. Here, we argue that the practise of kindness is by nature free and requires no other resources than the willingness of individuals. In health economics terms, the positive effects of kindness on mental health combined with a near-zero cost entail virtually no health loss, and an always-positive net health benefit and net monetary benefit[32]. Practising kindness is thus not only clinically effective, it is also cost-effective from a societal perspective.

## Social justice and autonomy

Third, Rose's population strategy has been criticised for focusing on health at the expense of other important societal values, such as justice or freedom[33]. However, such critiques can be avoided by identifying population strategies that are consistent with, rather than contrary to, important values in a liberal society. For example, social networks and support are recognised as social determinants of health, which the WHO defines as "non-medical factors that influence health outcomes."[34] While these non-medical factors can vary widely and range from "economic policies and systems, development agendas, social norms, social policies and political systems," our claim is that kindness can be one such part of a complex system of general, non-medical, social support for the population which is relatively accessible and low-cost. It has already been argued that preventive healthcare strategies which target social determinants of health should be prioritised "as a matter of social justice for all members of society and for the sustainability of quality healthcare."[35] We believe that addressing health inequities[36], through relatively costless and compassionate interventions, is a socially just way of exercising public health for all. As a matter of individual volition, kindness has the further advantage of posing little obstacles to freedom and autonomy, as it is by nature voluntary. This means that even if the practical complexities involved with disseminating kindness may in some contexts pose burdens and impositions upon some individuals, kindness in itself is neither unjust nor inconsistent with autonomy.

## Individual freedom and consent

Fourth, Rose's preventive strategy has been critiqued for imposing interventions on a nonconsenting population[33]. Such a critique can be accounted for in several ways. Many major public health interventions have been conducted on the starting premise that not all cases of overriding individual consent are necessarily morally wrong. Mask-wearing requirements in public spaces during a pandemic is an example of an intervention that may go against one's individual consent, for the sake of the public good, although this policy was controversial. A less controversial example may be the imposition of seat belt and motorcycle helmet laws: the law is coercive, yet the cost of foregoing consent is massively compensated by the effect of safety. We believe encouraging kindness is yet another uncontroversial example of a case where concerns about consent can be outweighed relative to the potential benefits. Further, it is fallacious to assume that the issue of consent is not already and inherently accounted for. Kindness – like love, generosity, compassion – is, in our view, only coherent when framed as a choice to be freely made. Kindness cannot be coercively extracted from individuals; if one does not consent to be kind or to practice and receive kindness, then we cannot talk about kindness in the relevant sense at all.

**Perspective**

## Empowering people

Fifth, Rose's approach has been criticised as stigmatising for labelling individuals and populations as "sick" to justify a mass prevention strategy[33]. Again, we argue that regardless of constructed labels such as "healthy" or "mentally ill", genders, age groups, sexual orientations, dietary preferences, occupations, political opinions, nationality, and so on, kindness can be of unconditional and universal benefit. This is especially because kindness is not only a preventive strategy; it may function as a proactive strategy of health promotion in general. As such, it does not need to be justified by reference to stigmatising or pathologising terms like "disease" and "ill-health"; the only requirement is the idea that health itself is in most cases a deeply vital good. From there, we can make sense of the concept of health promotion; that is, "the process of empowering people to increase control over their health and its determinants through health literacy efforts and multi-sectoral action to increase healthy behaviours."[37] While it may of course be the case that some groups benefit relatively more from kindness than others, this is all the more a reason to promote social interventions like kindness as necessary and worthwhile ways to improve health prospects.

## Do we need to wait longer?

On the practical aspect, there are numerous tools and techniques that are specifically designed to help generate kindness and compassion in individuals. For instance, in medical settings Mehta et al.[38] propose a self-kindness toolkit aimed at medical trainees[38]; in the context of loneliness the British Red Cross has created a self-kindness toolkit to improve wellbeing in adults[39]; and more universally, the Plum Village community founded by Nobel Peace Prize nominee Thích Nhất Hạnh – also known as the "father of mindfulness" – offers a free app including an extensive collection of guided meditations that help generate peace, calm, kindness and compassion towards oneself and others[40].

We have argued that practising kindness can be a cost-effective, socially just, inherently consensual, and empowering grassroots strategy of health promotion. Kindness can therefore surely complement other structural actions which target public health. Future academic studies can also certainly be useful to sharpen the terminology of "kindness", elucidate by which pathways kindness causes health benefits, assess whether practising kindness is superior/inferior to other interventions, or debate on whether the ground condition for practising kindness lies in the heart of human nature. Yet, in the meantime, practising kindness is already aligned with the principles of non-maleficence and beneficence, cost-effectiveness, social justice and consent, and is compatible with future decision-making and academic prospects. An emerging consciousness around its potential as a viable strategy to promote public health has already sparked initiatives pioneering a culture of kindness in medical care, such as: the *Excellence and Kindness in Research Training* (ELIS) in Denmark[41]; *The Kindness Coalition* at Stanford Medicine, US[42]; a call from the journal *BMJ Leader* upon medical leaders to place kindness at the centre of leadership action as essential to advancing the mission of healthcare systems[43]. Indeed, it may even be ethically unreasonable to postpone such a potentially powerful public health action that is already available[44].

To illustrate this ethical issue, we refer to the "parable of the arrow", well known in Buddhist traditions. To quote words of Zen peace activist Thích Nhất Hạnh[45]:

'Suppose a man is struck by a poisoned arrow and the doctor wishes to take out the arrow immediately. Suppose the man does not want the arrow removed until he knows who shot it, his age, his parents, and why he shot it. What would happen? If he were to wait until all these questions have been answered, the man might die first. Life is so short. It must not be spent in endless metaphysical speculation that does not bring us any closer to the truth.'

Future political actions and research studies on kindness will undoubtedly be valuable; meanwhile, we warmly invite our readers – whether we are patients, health care-givers, policy-makers, researchers, editors, or identifying ourselves with other labels – to practise kindness in our everyday lives, first with ourselves then with others. Kindness is immediately available to, for and within every one of us, in the present moment.

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

## Acknowledgements
We thank Dr. Tibor V. Varga and Dr. Adrian G. Zucco for their generous, constructive feedback on an earlier version of this manuscript. We thank Dr. Denise Utochkin and our many colleagues for inspiring us to practise kindness in our daily life at the office.

## Author Contributions

T.L. Nguyen contributed to this manuscript by bridging principles and learnings borrowed from Rose's approach to population health, empirical evidence of randomised controlled trials evaluating the effect of kindness, and Zen teachings on loving kindness and compassion. J.Y. Lee contributed to this manuscript by developing and consolidating the argumentation on medical ethics, social justice, consent, empowerment, and health promotion regarding the universal practice of kindness.

## Competing interests
The authors declare no competing interests.
