## [Peer Review file · Communications Medicine]

Kindness as a public health action

Corresponding Author: Dr Tri-Long Nguyen

Version 0:

Reviewer comments:

Reviewer #1

(Remarks to the Author)

Thank you for the opportunity to review this commentary, wherein the authors suggest that the universal practice of kindness is a potentially effective grassroots public health promotion action.

This manuscript was well-written, and I found it very interesting – it was truly a pleasure to read. The authors' fresh perspective on kindness as a health promotion approach was novel. I appreciated their comparison of kindness to infectious disease epidemiology and how individual action can have collective impact. Additionally, the authors' reference to kindness as a social determinant of health was noteworthy. I want to thank the authors for writing an article that was warm and inviting and left me feeling encouraged for the future of healthcare.

I believe the commentary will provide a valuable contribution to the literature and that it is ready for publication in its current form.

Reviewer #3

(Remarks to the Author)

Authors present kindness interventions for wellbeing. They attempt to analyze the role of kindness interventions to public health. However there are several deficiencies in therapy.

First kindness in clinical setting not defined. This is the same as promoting kindness as a social virtue? How can this be used in the medical/preventative setting?

Authors currently pointed out that kindness is embedded in medical ethics such as first to no harm. However, it is lack of application of these professional ethics that leads to many healthcare policies that are contrary to the concept of kindness, such as medical discrimination, lack of access to healthcare and healthcare disparities. I would appreciate if the authors could elaborate on the concept of kindness in practice of medicine and the politics of medicine.

On a global scale one observes that not only lack of kind acts but the enactment of cruel acts, who public health disasters. Recent example of this would be the war in Gaza, where medical profession has been surprisingly silent. I would appreciate if the authors could comment on the need for kindness in the international public health setting and especially in Gaza.

Overall, I believe that the authors are highly likely important issue, however it is somewhat naïve to link individual level kindness with public health, without addressing the inherent medical and social conflicts in the kindness paradigm.

Kindly revise the manuscript to incorporate these points.

Version 1:

Reviewer comments:

Reviewer #1

(Remarks to the Author)

I believe the manuscript is ready for publication in its current form. Thank you for the opportunity to review the revised version.

Reviewer #3

(Remarks to the Author)

22 January 2025

Dear Editor,

Manuscript ID: COMMSMED-24-0755

Title: *Kindness as a public health action*

We are pleased that both the Editorial Board and the reviewers have expressed their interest in publishing our article in *Communications Medicine*. We are very grateful to the two anonymous reviewers who generously commented on our original submission. Their constructive feedback helped us improve the quality of our manuscript, and we are more than thankful to them for this. We also want to express our gratitude to the Editor for the opportunity to revise our manuscript.

Please find enclosed a letter to the reviewers, in which we address all their comments, point by point. For the sake of readability:

- responses to reviewer comments are in blue;
- amendments to our initial manuscript are in red.

Specifically, as per request from the Editor, our revised manuscript now (1) defines kindness in the clinical setting and address how this can be implemented, (2) elaborates on the concept of kindness in the practice and politics of medicine, and (3) addresses the inherent medical and social conflicts in the kindness paradigm and the limitations of linking individual level kindness with public health.

The two authors approve this revised version of the manuscript. We declare no conflict of interest.

Sincerely,

T.L. Nguyen & J.Y. Lee

Department of Public Health, University of Copenhagen, Denmark

Reviewer #1

Thank you for the opportunity to review this commentary, wherein the authors suggest that the universal practice of kindness is a potentially effective grassroots public health promotion action.

This manuscript was well-written, and I found it very interesting – it was truly a pleasure to read. The authors' fresh perspective on kindness as a health promotion approach was novel. I appreciated their comparison of kindness to infectious disease epidemiology and how individual action can have collective impact. Additionally, the authors' reference to kindness as a social determinant of health was noteworthy. I want to thank the authors for writing an article that was warm and inviting and left me feeling encouraged for the future of healthcare.

I believe the commentary will provide a valuable contribution to the literature and that it is ready for publication in its current form.

Author's response: We warmly thank the Reviewer for their very positive feedback, which encourages us to pursue our work.

Reviewer #2

Authors present kindness interventions for wellbeing. They attempt to analyze the role of kindness interventions to public health. However there are several deficiencies in therapy.

- (1) First kindness in clinical setting not defined. This is the same as promoting kindness as a social virtue? How can this be used in the medical/preventative setting?

Author's response: We thank the Reviewer for their comment. This is true that a specific definition of kindness in clinical setting was lacking in our original submission. According to a recent review by Hake and Post (2023) on the definitions of kindness in healthcare, kindness can be broadly defined as *an action (mental or physical), that has a “warm” component, “directed toward fostering... well-being or flourishing”*.¹ While Hake and Post suggest defining kindness based on the beneficial result on *others*,¹ we underscore the need to extend the recipients of kindness: kindness towards others (patients and colleagues) must be conditional on kindness towards *oneself*. Indeed, echoing the Good Medical Practice guidance 2024 authored by the UK General Medical Council, while we believe that doctors must treat patients and colleagues with kindness (items 23 and 48), doctors should also take care of their own health and wellbeing as necessary conditions for their professional practice and seek advice from a suitably qualified professional if necessary (items 78 and 79).² We argue that it is fundamental to prevent kindness from being regarded as an “obligation” towards others (which, we argue in our paper, is by nature inconsistent with kindness), as this could result in a source of distress to doctors. This argument is akin to the fundamental distinction between *empathy* and *compassion*.^{3,4} Empathy is a mental construct that allows one to resonate with others feelings (positive or negative), which can result in a feeling of distress when observing others suffering. On the other hand, compassion is characterised by a feeling of warmth, concern, and care for the other, as well as a strong motivation to improve the other's wellbeing – which does not entail a feeling of distress. In this sense, we relate kindness to *compassion*, rather than empathy. Thus, we regard kindness as a path towards enhancing resilience of healthcare professions rather than merely a social virtue in the moral sense.

On the practical aspect, there are numerous tools and techniques that are specifically designed to help generate kindness and compassion in individuals. For instance, in medical settings Mehta et al. (2024) propose a self-kindness toolkit aimed at medical trainees ⁵ ; in the context of loneliness the British Red Cross has created a self-kindness toolkit to improve wellbeing in adults ⁶ ; and more universally, the Plum Village community founded by Nobel Peace Prize nominee Thích Nhất Hạnh offers a free app including an extensive collection of guided meditations that help generate peace, calm, kindness and compassion towards oneself and others.⁷

To address the important points raised by the Reviewer and incorporate these additional reflections, we have amended our manuscript as follows:

Page 3, first section:

“Borne out of understanding, love, and compassion

Kindness is a volitional endeavour, which does not necessarily require material gift, service or favour; it can be a grand gesture or a tender thought, a greeting or a smile, a “yes” or a “no”, to others and to ourselves. We refer to “kindness” to loosely designate thought, speech or action intended to benefit others,⁸ with no expectation of reward or recognition.⁹ We relate kindness to the Buddhist notion of *mettā* (Pali term for “benevolence” or “loving kindness”), the practice of which is borne out of *cultivating understanding, love, and compassion, first for ourselves and then for others*.¹⁰ While one may question how the genuine intention to benefit *others* can first be practised within *ourselves*, it seems reasonable to claim that one cannot offer to others what one does not have for oneself. This ~~sets a distinction between~~ ~~distinguishes~~ kindness ~~and from~~ people-pleasing, which comes from dependency, expects approval from others, and neglects understanding, love and compassion for ourselves.⁹

More specifically in the clinical setting, following a recent review by Hake and Post (2023) on the definitions of kindness in healthcare, kindness can be broadly defined as *an action (mental or physical), that has a “warm” component, “directed toward fostering... well-being or flourishing”*.¹ While Hake and Post suggest defining kindness based on the beneficial result on *others*,¹ we underscore the need to extend the recipients of kindness: kindness towards others (patients and colleagues) must be conditional on kindness towards *oneself*. We argue that it is fundamental to prevent kindness from being regarded as an “obligation” towards others, as this could result in a source of distress to doctors, thereby being inconsistent with the well-being and flourishing advocated by Hake and Post. Kindness is as much a *self*-relation as it is an orientation towards others. This argument is akin to the fundamental distinction between *empathy* and *compassion*.^{3,4} Empathy is a mental construct that allows one to resonate with others feelings (positive or negative), which can result in a feeling of distress when observing others suffering.³ It is possible for empathy to manifest as *pity* or even *desensitization*, which entails detachment and passivity.¹¹ On the other hand, compassion is characterised by a feeling of warmth, concern, and care for the other, as well as a strong motivation to improve the other’s wellbeing.³ Healthcare professionals may be vulnerable to “compassion fatigue”, given their proximity to those who are sick, suffering, and so forth.¹² In such cases, exercising *self*-compassion is a crucial precursor to showing kindness towards others; rather than being overcome with feelings of guilt or self-criticism, self-compassion would involve “acknowledging that suffering, failure, and inadequacies are part of the human condition, and that all people—oneself included—are worthy of compassion.¹³” In this sense, we relate kindness to *compassion*, rather than mere empathy. Thus, we regard kindness as a path towards enhancing resilience of public health rather than merely a social virtue in the moral sense.”

Page 7, last section:

“Do we need to wait longer?”

On the practical aspect, there are numerous tools and techniques that are specifically designed to help generate kindness and compassion in individuals. For instance, in medical settings Mehta et al. (2024) propose a self-kindness toolkit aimed at medical trainees ⁵ ; in the context of loneliness the British Red Cross has created a self-kindness toolkit to improve wellbeing in adults ⁶ ; and more universally, the Plum Village community founded by Nobel Peace Prize nominee Thích Nhất Hạnh – also known as the “father of mindfulness” – offers a free app including an extensive collection of guided meditations that help generate peace, calm, kindness and compassion towards oneself and others.⁷

We have argued that practising kindness can be a cost-effective, socially just, inherently consensual, and empowering grassroots strategy of health promotion. [...]

(2) Authors currently pointed out that kindness is embedded in medical ethics such as first to no harm. However, it is lack of application of these professional ethics that leads to many healthcare policies that are contrary to the concept of kindness, such as medical discrimination, lack of access to healthcare and healthcare disparities. I would appreciate if the authors could elaborate on the concept of kindness in practice of medicine and the politics of medicine.

Author's response: We thank the Reviewer for raising this concern. This is true that a culture of kindness and compassion is needed – mostly from the medical leadership – to support the practice and application of these professional ethics in healthcare systems. While the issues of medical discrimination, lack of access to healthcare and healthcare disparities still remain present, we would like to underscore that we do see an emerging culture of kindness and compassion in medical care. To cite a few examples, the Physical Medicine & Rehabilitation Research of Copenhagen (PMR-C) established across 3 university hospitals of the Capital Region of Denmark launched in 2024 the initiative *Excellence and Kindness in Research Training* (ELIS), which is endorsed by the Graduate School of Health and Medical Sciences at the University of Copenhagen and the Doctoral School in Medicine, Biomedical Science and Technology at the University of Aalborg.¹⁴ ELIS promotes training for the next generation of researchers and research leaders so that they can build and develop research groups that value and promote excellence and kindness together. At the same time, Stanford Medicine launched in 2024 *The Kindness Coalition* as a programme pioneering a culture of compassion in medical care.¹⁵ The same year, an editorial issued by BMJ Leader identified kindness as one of the strategic priorities of the journal, arguing that placing kindness at the centre of leadership action is essential to advancing the mission of healthcare systems.¹⁶ Meanwhile, the medical literature seems to reflect an emerging consciousness about kindness: a quick literature search in the PubMed database shows, for instance, that the number of articles matching the word “kindness” in 2022 was more than 10 times the one in 2007 (see figure below).

As for the “*first, do no harm*” principle, we meant that if kindness were harmful to individuals, then our argumentation about public health action would not hold. This principle is to be considered as with any medical intervention or medication. As the extensive literatures we cite throughout our paper show, kindness is consistently associated with an increase in well-being and no harm. In this sense, to quote Asch, Atkins, and Walling (2021), we do believe that “if kindness were a drug, the FDA would approve it”.¹⁷

In short, while we agree with the Reviewer that barriers to kindness are still prevalent, we strongly believe that recent initiatives and incentives, along with an emergence of the literature about kindness in medical care, can spark hope in implementing and sustaining a culture of kindness and compassion in healthcare systems.

To elaborate on these important points, we have amended our manuscript as follows:

Page 4:

“Evidence of clinical benefits

First, do no harm. This principle of *non-maleficence* is a central tenet of modern medical ethics, providing a standard that public health systems should uphold. Moreover, the principle of *beneficence* – which requires not only to prevent harm to individuals, but to positively benefit them – further outlines an ethical ideal for the provision of quality healthcare at the individual and population level. Thus, ~~It is clear that~~ if a public health intervention like practising kindness were harmful in some groups, then a population approach would elevate the risk in those groups and decrease not only the total effectiveness of the strategy,¹⁸ but violate the ethical standards of healthcare. Here, existing evidence consistently indicates the benefits and non-harms of practising kindness. When kindness takes the form of silent but caring intention, meta-analyses of randomised controlled trials showed that in comparison to no intervention, kindness-based meditation has a moderate effect in reducing depressive symptoms, anxiety and stress, and increases compassion and self-compassion, and positive emotions of kindness-practitioners.¹⁹⁻²² When kindness translates into action, a meta-analysis of 27 experimental studies shows that practising acts of kindness has a small-to-moderate effect on increasing well-being of the kindness-actors.⁸ In other words, empirical evidence shows that – whether it is in the form of caring thoughts or in gestures – practising kindness has at best moderate benefits for those who practise it, and at worst causes no harms, indicating that kindness as an intervention strategy is aligned with the principles of non-maleficence and beneficence. On the other hand, recent data shows that barriers to kindness, such as fears of compassion, are associated with mental health conditions²³; for instance, during the pandemic, fears of offering compassion to oneself/others and fears of receiving compassion from others were associated with

depression, anxiety and stress.²⁴ In short, empirical evidence about the benefit-to-harm balance consistently indicates a favourable effect of practising kindness; thus, as Asch, Atkins, and Walling (2021) write: “*if kindness were a drug, the FDA would approve it*”.¹⁷”

Page 7, last section:

“Do we need to wait longer?”

On the practical aspect, there are numerous tools and techniques that are specifically designed to help generate kindness and compassion in individuals. For instance, in medical settings Mehta et al. (2024) propose a self-kindness toolkit aimed at medical trainees⁵; in the context of loneliness the British Red Cross has created a self-kindness toolkit to improve wellbeing in adults⁶; and more universally, the Plum Village community founded by Nobel Peace Prize nominee Thích Nhất Hạnh – also known as the “father of mindfulness” – offers a free app including an extensive collection of guided meditations that help generate peace, calm, kindness and compassion towards oneself and others.⁷

We have argued that practising kindness can be a cost-effective, socially just, inherently consensual, and empowering grassroots strategy of health promotion. Kindness can therefore surely complement other structural actions which target public health. Future academic studies can also certainly be useful to sharpen the terminology of “kindness”, elucidate by which pathways kindness causes health benefits, assess whether practising kindness is superior/inferior to other interventions, or debate on whether the ground condition for practising kindness lies in the heart of human nature. Yet, *in the meantime*, practising kindness is already aligned with the principles of non-maleficence and beneficence, cost-effectiveness, social justice and consent, and is compatible with ~~these~~ future decision-making and academic prospects ~~and~~. An emerging consciousness around its potential as a viable strategy to promote public health has already sparked initiatives pioneering a culture of kindness in medical care, such as: the *Excellence and Kindness in Research Training* (ELIS) in Denmark¹⁴; *The Kindness Coalition* at Stanford Medicine, US¹⁵; a call from the journal *BMJ Leader* upon medical leaders to place kindness at the centre of leadership action as essential to advancing the mission of healthcare systems.¹⁶”

(3) On a global scale one observes that not only lack of kind acts but the enactment of cruel acts, who public health disasters. Recent example of this would be the war in Gaza, where medical profession has been surprisingly silent. I would appreciate if the authors could comment on the need for kindness in the international public health setting and especially in Gaza.

Overall, I believe that the authors are highly likely important issue, however it is somewhat naïve to link individual level kindness with public health, without addressing the inherent medical and social conflicts in the kindness paradigm.

Author's response: We thank the Reviewer for this honest comment. We totally agree with the Reviewer that there is a need for kindness in the international public health setting in response to enactment of cruel acts as during wars. Although we share some concerns with the Reviewer as to the silence of some communities around the war in Gaza (e.g. the academic community), we believe that among intellectuals, the medical community was at the forefront of denouncing the atrocities perpetrated against healthcare facilities and personnel, and civilians. For instance, in a press release on 15 October 2023, the World Medical Association called upon all parties involved in the conflict to:

- Respect established International Humanitarian Law (IHL) and not use health facilities as military quarters or depots, nor to target health personnel facilities and vehicles.
- Provide to health personnel the adequate conditions to treat all patients with humanity and in compliance with the ethical values of their professions, including medical neutrality.
- Use humanitarian corridors to permit the safe provision of health equipment and humanitarian supplies required in Gaza.²⁵

Locally in Denmark, this press release was echoed by a statement issued by the Danish Medical Association on 19 October 2023, asking the government to ensure a humanitarian corridor and evacuation of civilians.²⁶ On 29 May 2024, together with the World Health Organisation, the United Nations Relief and Works Agency (UNRWA), and the International Federation of Medical Students' Associations (IFMSA), The Lancet co-hosted a side event at the World Health Assembly on the health situation in Gaza (see: https://www.unrwa.org/sites/default/files/the_lancet-unrwa-who-ifmsa_side_meeting_wha77_29_may.pdf). The presence of IFMSA in this event was particularly timely and coincided with the mass student demonstrations that took place around the Western world in denunciation of the atrocities of the war. It is here important to notice that such solidarity movements were initiated at the individual level and quickly propagated at the international level. Indeed, there is a direct interconnection between individual action and collective action, as individuals are part of social networks. Drawing upon complex system thinking, recent models of diffusion on networked systems have allowed researchers to study the spread of diseases or behaviours, and the emergence of cascading phenomena.²⁷⁻³⁰ The implication is straightforward: whether it is about epidemics, student solidarity

actions, or peace movements, global phenomena can emerge from individuals by diffusion on networks, without necessarily the external intervention of so-called “structural” actions.

To make this connection between the individual and the collective clearer, and incorporate examples of solidarity movements in response to violent conflicts, we have amended our manuscript as follows:

Page 4:

“A mass approach to public health

To show how practising kindness can translate into an effective public health action, we use the influential framework of preventive medicine proposed by Rose, who argued that massively applying to the general population an intervention with small individual benefits could be more effective in reducing the number of disease cases than targeting only those at high-risk of disease.^{31 32} The rationale behind this “prevention paradox” is simple: although a preventive measure might be more effective in individuals at high risk, the proportion of those high-risk groups is often much lower than the low-risk population, such that targeting only those at high risk would result in preventing a lower number of absolute cases.^{31 32} Likewise, we argue that cultivating kindness, understanding, love and compassion, first for oneself, then for others – even if it had only small to moderate individual benefits – is not only for persons at high risk of or affected by mental health conditions: *the practice of kindness is for everyone.*

Further, assuming the practice of kindness can propagate across interconnected individuals within social networks (e.g., households, circles of friends, communities, etc.), both the exposure to and the effect of kindness could be amplified by a network multiplier, analogous to the reproduction number, R_0 , in infectious disease epidemiology.³³ This entails, like in epidemics, that any so-called “individual” action (e.g., vaccination) could in fact have a *collective* impact. **Indeed, from the spread of diseases to behaviours, recent models of diffusion on networked systems drawing upon complex system theory have allowed researchers to explain the emergence of cascading phenomena, from the individual to the collective.²⁷⁻³⁰ The implication is straightforward: whether it is about epidemics or solidarity movements, mass phenomena can emerge from individuals and have an impact on the collective. Examples of kindness and compassion actions emerging from individuals include historical and contemporary mass peace movements around the world in response to violence committed during wars.”**

References

1. Hake AB, Post SG. Kindness: Definitions and a pilot study for the development of a kindness scale in healthcare. *Plos One* 2023;18(7) doi: 10.1371/journal.pone.0288766
2. UK General Medical Council. Good Medical Practice, 2024.
3. Dowling T. Compassion does not fatigue! *Can Vet J* 2018;59(7):749-50. [published Online First: 2018/07/22]
4. Klimecki O., T. S. Empathic distress fatigue rather than compassion fatigue? Integrating findings from empathy research in psychology and social neuroscience. In: Oakley B., Knafo A., Madhavan G., et al., eds. *Pathological Altruism*. New York, New York: Oxford University Press 2011;pp. 1–23.
5. Mehta KK, Salam S, Hake A, et al. Cultivating compassion in medicine: a toolkit for medical students to improve self-kindness and enhance clinical care. *BMC Med Educ* 2024;24(1):291. doi: 10.1186/s12909-024-05270-z [published Online First: 2024/03/16 21:44]
6. British Red Cross. The self-kindness toolkit. [Available from: <https://www.redcross.org.uk/get-help/get-help-with-loneliness/wellbeing-support/self-kindness-toolkit>.
7. Plum Village Tradition of Zen Master Thích Nhất Hạnh. Plum Village App. [Available from: <https://web.plumvillage.app/>.
8. Curry OS, Rowland LA, Van Lissa CJ, et al. Happy to help? A systematic review and meta-analysis of the effects of performing acts of kindness on the well-being of the actor. *Journal of Experimental Social Psychology* 2018;76:320-29. doi: 10.1016/j.jesp.2018.02.014
9. Hasson G. *Kindness: Change Your Life and Make the World a Kinder Place*: Capstone 2018.
10. Thích Nhất Hạnh. *No Mud, No Lotus: The Art of Transforming Suffering*: Parallax Press 2014.
11. Zembylas M. The “Crisis of Pity” and the Radicalization of Solidarity: Toward Critical Pedagogies of Compassion. *Educational Studies* 2013;49(6):504-21. doi: 10.1080/00131946.2013.844148
12. Raab K. Mindfulness, Self-Compassion, and Empathy Among Health Care Professionals: A Review of the Literature. *Journal of Health Care Chaplaincy* 2014;20(3):95-108. doi: 10.1080/08854726.2014.913876
13. Neff KD. The Development and Validation of a Scale to Measure Self-Compassion. *Self and Identity* 2003;2(3):223-50. doi: 10.1080/15298860309027
14. Physical Medicine & Rehabilitation Research – Copenhagen (PMR-C). Excellence and Kindness in Research Training (ELIS). [Available from: <https://www.hvidovrehospital.dk/pmrc/pages/save-the-date.aspx>.
15. Stanford Medicine. The Kindness Coalition. [Available from: <https://medicine.stanford.edu/news/current-news/standard-news/the-Kindness-coalition.html>.
16. Klaber R, Mountford J, Allwood D. Kindness in healthcare: why it matters and why BMJ Leader will focus on it. *BMJ Lead* 2024;8(1):1-2. doi: 10.1136/leader-2024-000991 [published Online First: 2024/02/17]

17. Asch SM, Atkins DV, Walling A. If Kindness Were a Drug, the FDA Would Approve It. *J Gen Intern Med* 2021;36(2):263-64. doi: 10.1007/s11606-020-06343-7 [published Online First: 2020/11/28]
18. Adams J. When the population approach to prevention puts the health of individuals at risk. *International Journal of Epidemiology* 2004;34(1):40-43. doi: 10.1093/ije/dyh232
19. Galante J, Galante I, Bekkers M-J, et al. Effect of kindness-based meditation on health and well-being: A systematic review and meta-analysis. *Journal of Consulting and Clinical Psychology* 2014;82(6):1101-14. doi: 10.1037/a0037249
20. Lv J, Liu Q, Zeng X, et al. The effect of four Immeasurables meditations on depressive symptoms: A systematic review and meta-analysis. *Clinical Psychology Review* 2020;76 doi: 10.1016/j.cpr.2020.101814
21. Zheng Y, Yan L, Chen Y, et al. Effects of Loving-Kindness and Compassion Meditation on Anxiety: A Systematic Review and Meta-Analysis. *Mindfulness* 2023;14(5):1021-37. doi: 10.1007/s12671-023-02121-8
22. Zeng X, Chiu CPK, Wang R, et al. The effect of loving-kindness meditation on positive emotions: a meta-analytic review. *Frontiers in Psychology* 2015;6 doi: 10.3389/fpsyg.2015.01693
23. Merritt OA, Purdon CL. Scared of compassion: Fear of compassion in anxiety, mood, and non-clinical groups. *British Journal of Clinical Psychology* 2020;59(3):354-68. doi: 10.1111/bjc.12250
24. Matos M, McEwan K, Kanovský M, et al. Fears of compassion magnify the harmful effects of threat of COVID-19 on mental health and social safeness across 21 countries. *Clinical Psychology & Psychotherapy* 2021;28(6):1317-33. doi: 10.1002/cpp.2601
25. World Medical Association. The World Medical Association stands firmly for the principles of medical neutrality as defined by the Geneva Convention, and calls on all parties to respect International Law and the integrity of unrelated civilian populations. 2023 [Available from: <https://www.wma.net/news-post/the-world-medical-association-stands-firmly-for-the-principles-of-medical-neutrality-as-defined-by-the-geneva-convention-and-calls-on-all-parties-to-respect-international-law-and-the-integrity-of-unr/>].
26. Danish Medical Association. Læger: Regeringen bør presse på for overholdelse af den humanitære folkeret. 2023 [Available from: <https://laeger.dk/nyheder/2023-laeger-regeringen-boer-presse-paa-for-overholdelse-af-den-humanitaere-folkeret>].
27. Martín-Gutierrez S, Losada JC, Benito RM. Impact of individual actions on the collective response of social systems. *Sci Rep* 2020;10(1):12126. doi: 10.1038/s41598-020-69005-y [published Online First: 2020/07/24]
28. Masuda N, Porter MA, Lambiotte R. Random walks and diffusion on networks. *Physics Reports* 2017;716-717:1-58. doi: 10.1016/j.physrep.2017.07.007
29. Pastor-Satorras R, Castellano C, Van Mieghem P, et al. Epidemic processes in complex networks. *Reviews of Modern Physics* 2015;87(3):925-79. doi: 10.1103/RevModPhys.87.925

30. Zhang Z-K, Liu C, Zhan X-X, et al. Dynamics of information diffusion and its applications on complex networks. *Physics Reports* 2016;651:1-34. doi: 10.1016/j.physrep.2016.07.002
31. Rose G. Strategy of prevention: lessons from cardiovascular disease. *Bmj* 1981;282(6279):1847-51. doi: 10.1136/bmj.282.6279.1847
32. Rose G. Sick Individuals and Sick Populations. *International Journal of Epidemiology* 1985;14(1):32-38. doi: 10.1093/ije/14.1.32
33. VanderWeele TJ, Christakis NA. Network multipliers and public health. *International Journal of Epidemiology* 2019;48(4):1032-37. doi: 10.1093/ije/dyz010